# SPATIAL MATCHING LOSS FUNCTION FOR MASS SEGMENTATION ON WHOLE MAMMOGRAPHY IMAGES

## ABSTRACT

Breast cancer is one of the cancer types with a high mortality rate among women, and mammography is one of the primary means to improve the identification of breast cancer. Deep-learning-based approaches are among the pioneering methods for mass segmentation in mammography images; in this category of methods, the loss function is one of the core elements. Most of the proposed losses aim to measure pixel-level similarities. While the hard-coded location information is provided in these losses, they mostly neglect to consider higher-level information such as relative distance, sizes, and quantities, which are important for mass segmentation. Motivated by this observation, in this paper, we propose a framework for loss calculation in mass segmentation for mammography images that incorporates the higher-level spatial information in the loss by spatial matching between the prediction and the ground truth masks while calculating the loss. The proposed loss calculation framework is termed Spatial Matching (SM) loss. Instead of only calculating the loss over the entire masks that captures the similarity of the segmentation and the ground truth only at the pixel level, SM loss also compares the two in cells in a grid that enables the loss to measure higher-level similarities in the locations, sizes, and quantities. The grid size is selected according to each sample, which enables the method to consider the variation in mass sizes. In this study, Binary Cross Entropy (BCE) and Tversky are used as the core loss in experiments for the SM loss. AU-Net is selected as the baseline approach. We tested our method on the INbreast dataset. The results of our experiments show a significant boost in the performance of the baseline method while outperforming state-of-the-art mass segmentation methods.

## 1 INTRODUCTION

Breast cancer is reportedly one of the cancer types with a high mortality rate among women (Siegel et al., 2022). Thanks to powerful deep-learning-based approaches proposed in recent years for breast cancer identification, the performance of state-of-the-art research in this field has been immensely boosted. This is specifically evident for mammography, as one of the most common screening tools with reported effectiveness in reducing mortality rate (Nyström et al., 2002). In practice, automated breast cancer detection could be used in the clinical setting to reduce the cost of a second reader (Batchu et al., 2021; McKinney et al., 2020).

While mass detection approaches Liu et al. (2021b;a) provide the bounding boxes of the mass, mass segmentation provides boundary information in addition to location and size, which is important for determining the abnormality type Guliato et al. (2007); Sun et al. (2020); Lee et al. (2017). Some of the challenges for mass segmentation are diversity of appearance and sizes in the abnormalities and pixel class imbalance. In this study, we propose a spatial matching loss framework that incorporates higher-level information such as distance, size, and quantity in the calculation of a given core loss, aiming to alleviate the limitations of the previously proposed loss functions. This is specifically important for mass segmentation as spatial matching compares the masks locally and could alleviate the pixel class imbalance problem, which is a major problem in mass segmentation. In addition, due to the variety of sizes and appearance of the masses, specifically for samples with higher ACR density, distinguishing the mass from other regions of the image is more challenging, and the total mismatch between the location, quantity, and size of predicted and true masses is likely even in recent well-developed methods. Therefore, the proposed loss is beneficial for handling these challenges, which are common in mass segmentation in mammography images.

Figure 1: Examples to illustrate the challenges for BCE loss. The pink and green shapes are true and predicted masses, respectively. In each pair of images, the loss value is the same. Demonstrated challenges are: (a) Ignoring the quantity of predicted masses with respect to true mass, (b) the bias of the loss to correctly predict the majority class and ignoring the distance of the predicted and true masses, (c) indifference to the false positive and false negative rates.

Widely used loss functions mostly target the pixel-level matching of the ground truth and prediction masks. For instance, Binary Cross Entropy (BCE) (Yi-de et al., 2004) is among the widely used loss functions for mass segmentation. Figure 1 summarizes the most challenging properties of the BCE loss for mass segmentation. In Figure 1, simplified examples of overlaid ground truth and prediction masks are presented. The gray color indicates that the pixels in both of the masks belong to the background class. The pink and green color shapes present the masses in the ground truth and the prediction mask, respectively. Each of the pairs of images in a, b, and c in Figure 1 presents a challenge for BCE. The value of the BCE loss and the total size of the green and pink regions in the left and right images are the same in each pair. In Figure1a, BCE is not able to differentiate between multiple predicted masses (left) versus only one (right) when the overall sizes remain the same. In Figure 1b, the BCE loss value is the same in both images, showing that BCE neglects to consider the distance between the real and predicted masses. In addition, it shows that BCE will be biased toward correctly predicting the background class as both predicted and the ground truth masses are very small compared to the background. Finally, Figure 1c shows that BCE does not differentiate between the different sizes for false positives and false negatives.

Several studies have attempted to address the bias of the BCE through weighting strategies to balance the impact of each class that could compensate for the pixel class imbalance problem in the segmentation task for medical images. However, they mostly fall short of moving beyond pixel-level comparison of the ground truth and the prediction and only focus on the class imbalance problem. Tversky loss (Salehi et al., 2017) is one of the losses that eliminates the correctly predicted background (easy samples) from the calculation of the loss and also weights the false positive and false negative terms differently to differentiate the contribution of each in the loss. However, the weights are fixed hyper-parameters regardless of the severity of the pixel class imbalance problem. Moreover, it fails to move beyond pixel-level comparison and to take the distance and quantities into consideration. Unlike previous attempts to address the aforementioned challenges, in this paper, instead of only considering a pixel-level solution that could address the problem partially, we propose the spatial matching loss framework, which incorporates the idea of local comparison for the masses in a grid to embed the regional information in a given core loss. This, in turn, enables our approach to use the distance, size, and quantity information in the calculation of the loss. All the aforementioned proposed strategies are possible with the concept of spatial matching, which was inspired by the usage of Spatial Pyramid Matching (SPM) (Lazebnik et al., 2009) in (Alinia & Razzaghi, 2017; Aliniya & Razzaghi, 2018) for scene parsing. It should be noted that the SM loss could be used for other segmentation tasks for medical images with similar properties and could also be extended to other domains for binary segmentation tasks. We tested the proposed SM loss with BCE and Tversky losses as the core losses on AU-Net, one of the state-of-the-art methods that are proposed specifically for mass segmentation in mammography images. The experimental results show that the SM loss significantly outperforms state-of-the-art methods.

The following is the summary of contributions of the paper: 1) Proposing the spatial matching loss for mass segmentation that allows the pixel-level losses to utilize spatial correspondences at the local and global levels. 2) Introducing the gird-base loss calculation for local comparison. 3) Sample-level cell size extraction that improves the local comparison by incorporating the size of the true mass in the generation of the grid. 4) Introducing two ways to utilize the information provided by the grid to incorporate the size, distance, and quantity of the masses. 5) Evaluating the proposed method and conducting ablation studies on INbreast, which is a benchmark dataset for mass segmentation

in mammography. 6) Quantitatively analyzing and comparing the findings from our experimental results for the proposed method with state-of-the-art approaches.

## 2 RELATED WORK

This section is dedicated to reviewing the related work to this study from two perspectives. Firstly, a brief review of recent deep-learning-based approaches for mass segmentation on whole-view mammography images (the core task of this paper) is presented. Secondly, current loss functions that are proposed mainly for binary segmentation tasks are reviewed.

### 2.1 MASS SEGMENTATION ON WHOLE MAMMOGRAMS

In this section, only deep-learning-based mass segmentation methods for whole mammography images have been reviewed. The ROI-based approaches have been excluded due to the differences in their challenges and strategies for solving them compared to methods for the whole mammogram.

Inspired by the fully convolutional architecture in (Long et al., 2015), Ronneberger et al. (2015) proposed U-Net which is one of the early deep-learning-based segmentation methods for medical images. U-Net is designed with a symmetric encoder-decoder architecture that performs well for limited data. This property of U-Net makes it specifically favorable for biomedical image segmentation tasks with limited datasets. U-Net combines low-level location information from the encoder with high-level contextual information from the decoder.

Inspired by the effectiveness of U-Net, a new wave of modification and tailoring of U-Net for different medical tasks has emerged (Wu et al., 2019; Zhang et al., 2023; Zhou et al., 2018; Li et al., 2020; Cao et al., 2023; Oktay et al., 2018; Song et al., 2019; Hai et al., 2019), continuing to push the performance boundaries of medical image segmentation. In this category, in (Li et al., 2019), authors proposed a U-Net-based approach based on the idea of utilizing a densely connected network in the encoder and a CNN with attention gates in the decoder. Another line of research within the scope of multi-scale studies is (Chen et al., 2020), where the generator was designed as an improved version of U-Net. Before sending the segmentation results to the discriminator, multiscale results were created for three critics with different scales in the discriminator. Rajalakshmi et al. (2021), developed an approach employing the error of the outputs of intermediate layers relative to the ground truth labels as a supervision signal to boost the model's performance.

In (Sun et al., 2020), the authors introduced an attention-guided dense-up-sampling asymmetric encoder-decoder network (AU-Net) with an intermediate up-sampling block, which includes a channel-wise attention mechanism designed to leverage the beneficial information presented in both low and high-level features. To mitigate the problem of relatively low performance of the U-Net approach on small-size masses, Xu et al. (2022) proposed to use a selective receptive field module with two parts, one for generating several receptive fields with different sizes and one for selecting the appropriate size of the receptive fields. AU-Net has been chosen as the baseline in this study.

### 2.2 LOSS FUNCTIONS FOR MEDICAL IMAGE SEGMENTATION

This section reviews the loss functions widely used for the binary segmentation task. Loss functions could be categorized into two groups with respect to the pixel dependencies: loss functions that consider the pixels as independent entities or those that take the dependencies between the pixels into consideration. In the pixel-level category, BCE (Yi-de et al., 2004) penalizes the discrepancy between predicted and ground truth for all pixels. As improved versions of BCE, Weighted Binary Cross Entropy (Pihur et al., 2007) and Balanced Cross Entropy (Xie & Tu, 2015) differentiate between the effect of false positives and false negatives through weighting coefficients. With a similar goal, Focal loss (Yeung et al., 2022) further improved BCE by changing the magnitude of the loss according to the hardness of the examples based on the prediction confidence for them. Entirely eliminating the true negative, Dice loss (Milletari et al., 2016) addressed the pixel class imbalance problem (Jadon, 2020), formulated as the ratio of correctly classified pixels to the total number of positive pixels in the prediction and ground truth masks. Tversky loss (Salehi et al., 2017) provided a way to control the contribution of the false positive and the false negative terms by weighting these terms (they could have different weights).

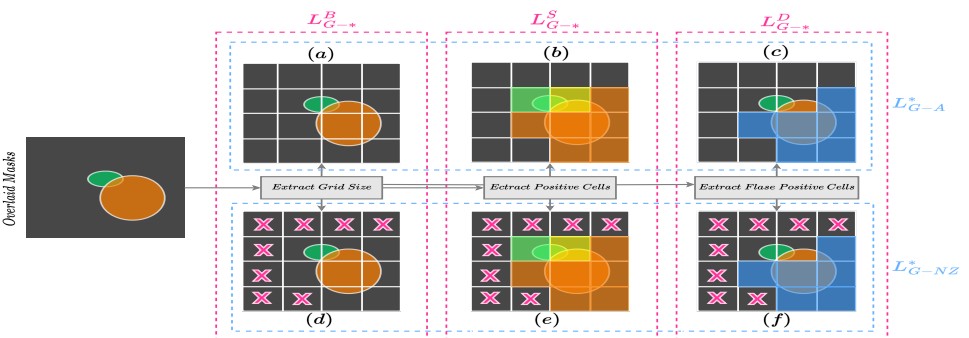

Figure 2: Overview of the grid loss in the proposed method. The green oval shape and the orange circle are true and the predicted masses, respectively. (a-c) are basic, scale, and distnce-based SM losses for $L_{G-A}^*$, respectively. (e-f) present the same variations of $L_{G-NZ}^*$, in which the cells containing only true negatives are eliminated. Considering cells with positive pixels (true or predicted) for the scale-based version (b,e), the green, orange, and yellow show cells containing TP (TP and FP) and FP, respectively. For the distance-based version (c, f), blue cells contain FP.

All the aforementioned losses are pixel-level (i.e., they consider the pixels as independent entities). While being effective, they neglect to consider the relationship among pixels in the loss calculation. In this category, inspired by the influence of the structural term in the SSIM (Wang et al., 2004) used in (Huang et al., 2020), the authors of Structural Similarity Loss (SSL) (Zhao et al., 2019b) proposed to weight the cross-entropy of every two pixels based on the error between two image regions, which indicates the degree of linear correlation between them. The authors also proposed to eliminate pixels with low error and emphasize on those with high error by thresholding the error rate. RMI (Zhao et al., 2019a) is another research aiming to incorporate structural similarity by converting regions around a center (pixel) to a multi-dimensional point and then maximizing the MI between multidimensional distributions.

Several compound losses (Yeung et al., 2022; Salehi et al., 2017; Taghanaki et al., 2019) have been proposed to reap the benefits of different losses by combining two or more of them in a weighted sum. Combo loss (Taghanaki et al., 2019) was proposed to control the contribution of false positive and false negative terms in the BCE by weighting them differently, where the total loss is a weighted sum of BCE and Dice loss. Adaptive sample-level prioritizing (ASP) (Aliniya et al., 2023a) and Density-ASP (Aliniya et al., 2023b) losses are two approaches for weighting the loss terms adaptively, according to each sample. The loss is calculated in an adaptive manner by controlling the influence of each loss term according to each sample. While ASP loss uses the ratio of the mass to image size as a re-weighting signal, Density-ASP leverages the density category associated with the image sample as the re-weighting signal. In addition, instead of re-weighting the Dice and BCE losses in the hybrid setting, Density-ASP moves beyond the pixel-level losses by re-weighting region-level and pixel-level loss terms.

All aforementioned losses could be categorized into two groups: region-level and pixel-level losses. The proposed SM loss could be integrated with both types of losses to boost their performances. It should be noted that our spatial matching framework while being similar to region-level losses (RMI, SSL, and Density-ASP losses) in the respect that all of them try to incorporate the spatial information in the calculation for the loss, is greatly different. Region-level losses aim to consider the value of the surrounding pixels directly in the calculation of the loss for a centering pixel. However, the SM is formulated in a way that compares the global and local spatial similarities between the images. The following section is dedicated to presenting the proposed method in detail.

## 3 PROPOSED METHOD

The proposed method is presented in Algorithm 1. The main idea of $SM$ loss is to provide a framework for loss calculation that incorporates higher-level spatial information in a given core loss $L$ (in this study, BCE and Tversky losses presented in Equations 2 and 3 have been selected as

core losses to evaluate SM loss). To achieve this goal, given prediction ($P$) and ground truth ($Y$) masks and a grid $G = \{c_1, ..., c_{N_G}\}$ over both masks ($N_G$ is the number of the cells), we propose to calculate the loss over the cells ( presented in Figure 2), in a grid in addition to the loss for whole masks as presented in Equation 1.

---

**Algorithm 1:** Spatial Matching Loss

**Data:** $P$, $Y$, $Grid$, $G$, $L$, $SM$
**Result:** $L_{SM}$

1   $L_I \leftarrow$ calculate $L$ for the whole masks;
2   **foreach** $i \in Grid$ **do**
3      $y_i, p_i \leftarrow$ get $i$-th cells in $Y$ and $P$;
4      $L_i \leftarrow$ compute $L$ over $p_i$ and $y_i$;
5      **if** $SM$ *is Basic* **then**
6         $L_G \leftarrow L_G + L_i$;
7      **else**
8         **if** $SM$ *is Scale or Scale-Distance* **then**
9            $NZ_P \leftarrow NZ_P + 1$ if $p_i$ contains mass;
10            $NZ_Y \leftarrow NZ_Y + 1$ if $y_i$ contains mass;
11            $L_G \leftarrow L_G + L_i$;
12         **if** $SM$ *is Distance or Scale-Distance* **then**
13            $D_i \leftarrow$ distance from $p_i$ to closest mass in $Y$ if $p_i$ contains mass and $y_i$ does not contain mass;
14            $L_G \leftarrow L_G + L_i \times \sigma(D_i)$;

15   **if** $G$ *is G-A* **then**
16      $L_G \leftarrow$ Divide $L_G$ by number of cells.
17   **else**
18      $L_G \leftarrow$ Divide $L_G$ by the number of cells containing at least one of the FP, FN, or TP.
19   **if** $SM$ *is Basic or Distance* **then**
20      $L_{SM} \leftarrow L_I + L_G$;
21   **else**
22      $L_{SM} \leftarrow L_I + L_G \times \sigma(NZ_P/NZ_Y)$;

---

$$L_{SM} = L_I + L^*_{G-*} \tag{1}$$

$L_I$ is the loss over the entire masks and $L^*_{G-*}$ presents the grid loss. We propose two different ways to aggregate the losses for cells $L_i$ (loss value for $i_{th}$ cell) into the grid loss $L_G$ (the name of which will replace the superscript star) and three versions for calculating the $L_G$ (the name for these will replace the star in subscript). Incorporating the grid loss in the calculation of the $SM$ loss allows the method to capture the local spatial information in addition to the overall pixel-level loss, as the distribution of the matches and mismatches will differ at the local level. For instance, cells only containing true negatives in both masks will be ignored from the total loss calculation, reducing the loss's bias toward correctly predicting the majority class. Furthermore, one important aspect of the $SM$ loss is the selection of the height and width of the cells, which plays a crucial role in the performance of the method. We propose to generate the grid according to each sample in which the height and width of the cells are selected based on the size of the masses in $Y$ (In Figure 2 the green oval and orange circle are the true and predicted masses respectively). To this end, the minimum height and width of the bounding boxes surrounding the masses in $Y$ have been used. To ensure equal division of the masks, the numbers have been rounded to the closest number in an exponential sequence (up to 256, which is the image size) of numbers that are powers of 2 for the size of the grid. This has been depicted in the "Extract Grid Size" module in Figure 2. Using the mass size for determining the size of the grids allows the method to reduce the impact of the class imbalance problem by customizing the size according to the mass size (minority class).

$$BCE = -\frac{1}{N}\Big(Y_j log(P_j) + (1 - Y_j)log(1 - P_j)\Big) \tag{2}$$

$$T = \frac{\sum_j y_j p_j}{\sum_j y_j p_j + \alpha \sum_j y_j (1 - p_j) + \beta \sum_j (1 - y_j) p_j} \qquad (3)$$

### 3.0.1 AGGREGATING THE CELL LOSSES INTO THE GRID LOSS

We propose two ways to combine the cell losses to form the grid loss. One is averaging loss values over all of the cells (Equation 4), which is called $L^*_{G-A}$, as shown in Figure 2(a-c).

$$L^*_{G-A} = \frac{1}{N_G} \sum_i^{N_G} L_i \qquad (4)$$

In the second variation, we aim to target the class imbalance problem by eliminating the easy cells that only contain true negative samples. This variation (Figure 2(d-f)), $L^*_{G-NZ}$ is presented in Equation 5 in which $1\{\}$ returns one only if the condition inside $\{\}$ is met; otherwise, it will be zero. For the condition, we used summation to specify cells containing at least one of the true or false positives will be considered. As shown in Figure 2(d-f), all of the cells containing only true negative (shown by cross sign) have been eliminated.

$$L^*_{G-NZ} = \frac{\sum_i^{N_G} L_i}{\sum_i^{N_G} 1\{\sum_k y_i^k + \sum_k p_i^k > 0\}} \qquad (5)$$

Here, $y_i$ and $p_i$ are regions from $Y$ and $P$ in $i_{th}$ cell and $k$ presents $k_{th}$ pixel in the region $i$ .

### 3.0.2 STRATEGIES FOR CALCULATING THE GRID LOSS

Up to this point, the basic idea of the SM loss has been presented by simply calculating the loss for each cell $L_i$ in the grid (Equation 6) called basic SM loss (Figure 2(a,d)). This variation captures local similarities in the masks. In subsection 3.0.1, the basic SM loss was used for explanation.

$$L^B_{G-*} = \sum_i^{N_G} L_i \qquad (6)$$

The SM loss could be further improved by incorporating the scale of the real and predicted masses to overcome the size differences and also the differences in the quantities of the masses. This variation, called scale SM loss, is presented in Equation 7, in which the $L_G$ will be re-scaled with the scale ratio (Equation 8). The ratio is defined as the number of the cells containing the positive pixels in the predicted mask (shown by orange and yellow color cells in Figure2(b,e)) over the number of the cells containing the positive pixels in the ground truth ( shown by green and yellow colors in Figure 2(b,e). Yellow color presents cells that contain both predicted and true masses. The "Extract Positive Cells" module identifies these cells.

$$L^S_{G-*} = \sigma(S) \sum_i^{N_G} L_i \qquad (7)$$

$$S = \frac{\sum_i^{N_G} 1\{\sum_k p_i^k > 0\}}{\sum_i^{N_G} 1\{\sum_k y_i^k > 0\}} \qquad (8)$$

The $\sigma$ is a sigmoid function used to keep the ratio between 0 and 1, in addition to adding some non-linearity to control the effect of the scale ratio. The numerator measures how large the predicted mass is relative to the true mass as the cell size is defined according to the true mass size, penalizing the large differences by increasing the loss. The cell size is a function of true mask size, so the denominator will be four at maximum for samples with one mass, which is the majority of samples. The values of more than one occur if the mass is divided between the cells. The sigmoid function

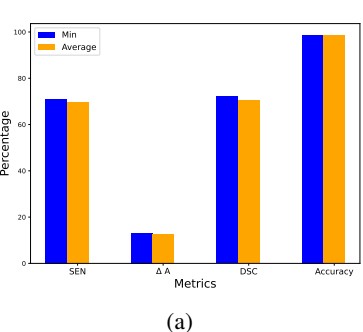 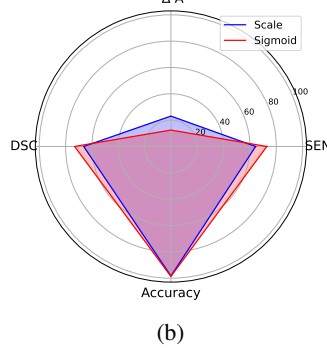

(a)  (b)

Figure 3: Results of ablation studies for (a) using minimum and average of the real masses sizes for determining the size of cells, (b) using sigmoid versus direct values of the scale and distance.

helps to handle the small changes that can present small noise, like division up to four in the masses. One situation in which the denominator can have large values is when there are several masses in one sample. For these cases, as the cell size is selected based on the size of the smaller mass, the scale-based loss would not accurately present the initial idea of the scale, depending on the sizes of the true and predicted mass. The denominator might be considerably bigger than the numerator, which will decrease the impact of the grid-based loss depending on the magnitude of these samples, which is appropriate as the cell loss does not accurately present the true mass size. The scale implicitly penalizes the larger qualities of the predicted mass compared to the real mass, considering that if a certain number of predictions for the positive class are presented in one cell, they all will be counted as one cell. However, if the same number of positively predicted pixels are presented in five different parts of the image (more quantities), the number of cells for the same number of pixels will be larger; therefore, SM will penalize the difference in the quantity to some extent compared to pixel-level losses. It should noted the ratio of the number of pixels instead of the cells in the scale would not incorporate the quantities into the calculation.

The last variation of the SM loss presented in Equation 9 aims to incorporate the distances between the real and predicted masses in the calculations. To this end, for all cells that contain only false positive and true negative (shown by blue color in Figure 2(c,f)), their loss value will be rescaled by the distance of the cell to the closest real mass to penalize far predictions more. The "Extract False Positive Cells" module identifies these cells

$$L_{G-*}^{D} = \sum_{i}^{N_G} \sigma(d_i) L_i \tag{9}$$

The $\sigma$ function in Equation 9 is used for the same purpose as in Equation 7. Finally, the scale and distance-based SM loss could be used in combination. The overview of the approach is presented in more detail in Algorithm 1. The inputs are the ground truth ($Y$), prediction ($P$), aggregation type ($G$), loss type (BCE or Tversky), grid, and SM loss variation ($SM$). The output is the SM loss.

## 4  EXPERIMENTAL RESULTS

### 4.1  DATASET AND EVALUATION METRICS

The INbreast dataset contains 410 images, including various abnormality types. Only 107 of the images containing masses have been used in this study. A 5-fold cross-validation was employed, which is a commonly used setting for the measurement of the performances of methods on the INbreast dataset. The dataset was randomly divided into training (80%), validation (10%), and test (10%) sets. We have normalized the intensity of the images, and all images have been resized to $256 \times 256$. No data augmentation or image enhancement was considered in our experiments in order to only measure the effect of the loss function. For AU-Net (baseline), the batch size was set to four; the learning rate was set to 0.0001.

| | Metrics | $L^*_{G-A}$ | | | | | $L^*_{G-NZ}$ | | | | |
|---|---|---|---|---|---|---|---|---|---|---|---|
| | | DSC | {ΔA} | SEN | HAU | ACCU | DSC | {ΔA} | SEN | HAU | ACCU |
| BCE | SM-B | 72.12 | 13.09 | 70.88 | 5.00 | 98.69 | 71.25 | 16.21 | **71.95** | 5.08 | 98.54 |
| | SM-D | 73.13 | 12.45 | 72.80 | 5.00 | 98.66 | **73.03** | 14.77 | 70.89 | **4.97** | 98.67 |
| | SM-S | 73.03 | 14.44 | **74.42** | 5.11 | 98.60 | 72.76 | 16.53 | 71.16 | 5.01 | **99.32** |
| | SM-SD | **75.70** | **10.29** | 73.79 | **4.89** | **99.46** | 71.12 | **12.62** | 66.95 | 5.14 | 98.59 |
| Tversky | SM-B | 71.77 | 14.88 | 70.70 | 5.06 | 98.58 | **75.92** | **11.85** | **78.41** | 5.00 | 98.72 |
| | SM-D | 72.25 | 17.58 | 74.92 | 5.11 | **99.17** | 71.86 | 13.51 | 70.65 | 5.06 | 98.57 |
| | SM-S | **77.24** | 09.49 | **79.38** | **4.84** | 98.78 | 73.38 | 12.31 | 75.82 | **4.96** | **99.16** |
| | SM-SD | 72.72 | **05.80** | 72.87 | 4.97 | 98.57 | 70.36 | 21.34 | 69.46 | 5.03 | 98.55 |

Table 1: Comparison of different variations of the proposed method using BCE and Tversky losses.

| Method | DSC | ΔA | SEN | SEN | Accuracy |
|---|---|---|---|---|---|
| ARF-Net | 70.05 | 30.05 | 59.59 | - | 98.71 |
| AU-Net | 65.32 | 23.68 | 57.95 | 5.29 | 98.46 |
| ASP-Q | 68.03 | 25.04 | 63.12 | - | 98.54 |
| ASP-L | 71.92 | 22.31 | 64.56 | - | 98.71 |
| ASP-C | 74.18 | 19.28 | 67.21 | - | 98.78 |
| Density-ASP | 74.59 | 10.91 | 78.16 | - | 98.65 |
| **SM-SD(BCE)** | 75.70 | 10.29 | 73.49 | 4.89 | **99.46** |
| **SM-S(Tversky)** | **77.24** | **09.49** | **79.38** | **4.84** | 98.78 |

Table 2: Comparison of the proposed method with state-of-the-art approaches.

Due to the pixel class imbalance problem associated with mammograms, we have selected several metrics to better illustrate the strengths and weaknesses of the methods. Specifically, the Dice Similarity Coefficient (DSC), Relative Area Difference ($\Delta A$), Sensitivity, Hausdorff Distance (HAU), and Accuracy (ACCU) have been selected due to the complementary information they provide for the evaluation of methods. For the AU-Net method, the publicly available implementation was used, and we implemented the ARF-Net approach according to the description in the paper.

### 4.1.1 ABLATION STUDY

In this section, the results of experiments for selecting various parameters that influence the performance of the SM loss have been presented. Table 1 summarises the results for each of the BCE and Tversky losses using all the variations of the SM loss. As shown in Table 1, the SM loss significantly improves upon the BCE loss across all of the metrics. SM loss using the BCE performs better in the $L_{G-A}$ version, and the best performance is achieved for the combination of scale and distance.

Regarding the Tversky loss, SM loss provides a robust improvement across all the metrics. SM loss using Tversky as the core loss also yields better results in $L_{G-A}$ setting in general. Also, the best performance of the SM loss is achieved by the $L^S_{G-A}$ using the Tversky loss (SM-S row in the $L^*_{G-A}$ for Tversky) section. While achieving overall improvement on both BCE and Tversky losses, the relative performance of the basic, scale, and distance-based SM loss vary in different settings; we attribute this to the fact that some of them may target the same challenge at the same time. For example, for the BCE in the $L^*_{G-A}$, the only varying factors are the effect of basic, scale, and distance-based and a combination of scale and distance-base versions. Appendix A provides the result for Dice loss as the core loss.

In addition to the previous experiments, we conducted several experiments on the impact of the selection of the grid size and the re-scaling method for the distance and scale-based SM loss. Firstly, as some mammography images contain several masses with different sizes when computing the grid size, the question is whether to use the average of the sizes or the minimum size. As the maximum will increase the effect of the true negative, it is not considered in this experiment. Figure 3a depicts the results of the experiments, highlighting the advantage of using minimum value. It should be noted that a minority of images in the mammography dataset contain several masses. Therefore, given the boost in the result, this is still an important factor to consider. In the second experiment, we explored the idea of using sigmoid (that adds some non-linearity toward the tails of the ratios

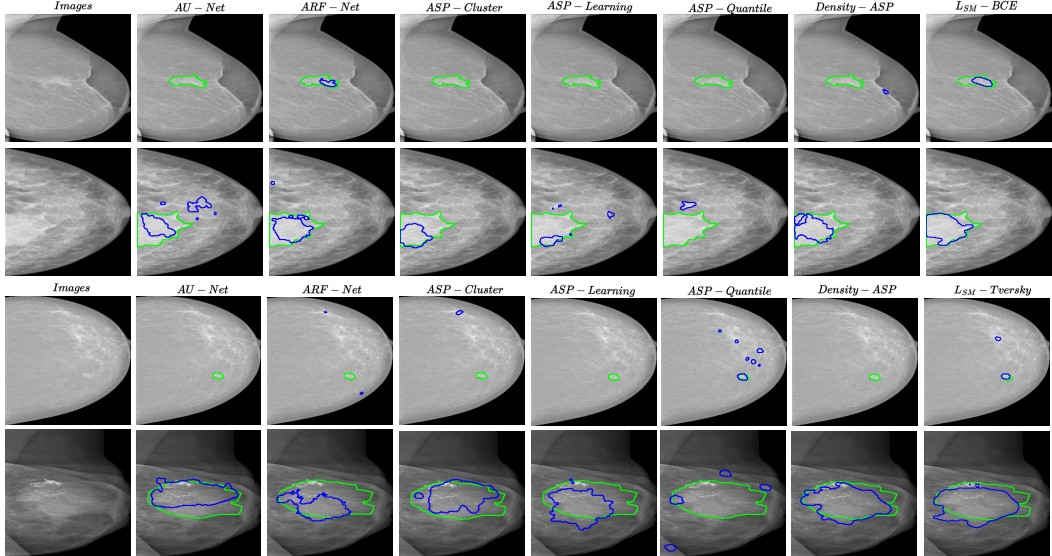

Figure 4: Examples of results for AU-Net, ARF-Net, ASP variations, Density-ASP, and SM loss (two top rows are for BCE loss, and the rest are for the Tversky loss). The green and blue lines are the borders of ground truth and prediction masks, respectively.

for scale and distance) and using the ratios directly, in which even small changes in the ratio will be reflected in the loss. As shown in Figure 3b the sigmoid performs better compared to directly including the ratios. One reason could be that directly using ratio might add noise to the training.

### 4.1.2 COMPARISON WITH STATE-OF-THE-ART METHODS

Table 2 presents the comparison between the best-performing version for the SM loss and the state-of-the-art methods. AU-Net and ARF-Net use hybrid loss, which is a weighted sum of Dice and BCE losses. ASP loss variations and Density-ASP loss are also hybrid losses with sample-based weighting strategies, and they utilize the mass size and density categories of the images in the calculation of the loss, respectively. As shown in Table 2, SM loss significantly outperformed all of them. Compared to the baseline method the improvements are as follows: ($DSC : +11.92, \Delta A : -14.19, SEN : +21.43, HAU : -0.45, Accuracy : +0.32$). ARF-Net has a different architecture, which incorporates the size in the design; also, sample-level losses use additional information alongside the ground truth segmentation. Compared to these methods, SM loss does not use additional information or the sizes in the architecture (AU-Net) yet outperformed them by a considerable margin. Some of the segmentation examples for all the methods have been presented in Figure 4, in which SM loss improves the segmentation results for various mass sizes.

## 5 CONCLUSION

In summary, in this paper, we present a new framework for calculating the loss for mass segmentation in mammography images. With the goal of addressing the common challenges in this domain, including pixel class imbalance, various sizes, and distance between the real and predicted masses, the proposed spatial matching loss leverages the idea of a sample-level grid that provides the commonly used losses with the ability to incorporate local spatial similarities between the masks. For the grid size, we propose to use the size of the real masses, which enables the method to reduce the pixel class imbalance problem. In addition, the grid information enables the SM loss to capture the differences between the size, quantity, and location of the masses. The proposed SM loss is evaluated using BCE and Tversky as the core losses on the INbreast dataset which is a benchmark dataset for mass segmentation. The experimental results confirm the promise of the SM loss by robustly outperforming the state-of-the-art methods.

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

| | $L^*_{G-A}$ | | | | | $L^*_{G-NZ}$ | | | | |
|---|---|---|---|---|---|---|---|---|---|---|
| | DSC | $\{\Delta A\}$ | SEN | HAU | ACCU | DSC | $\{\Delta A\}$ | SEN | HAU | ACCU |
| SM-B | 70.76 | **07.76** | 68.13 | 5.08 | 98.52 | 73.05 | 22.42 | 72.03 | **4.98** | 98.66 |
| SM-D | 73.17 | 11.82 | 71.69 | 5.15 | 98.65 | **75.31** | 11.25 | **76.49** | 5.02 | **98.72** |
| SM-S | 73.17 | 13.13 | 68.86 | **4.92** | 98.72 | 73.86 | **07.66** | 74.78 | 5.03 | 98.66 |
| SM-SD | **73.72** | 13.18 | **72.58** | 5.09 | **99.41** | 72.11 | 09.08 | 70.57 | 4.98 | 98.59 |

Table 3: Results for SM loss with dice as the core loss.

## 6 APPENDIX A: DICE LOSS AS CORE LOSS

Dice loss is one of the losses that is commonly used in combination with BCE loss for mass segmentation. We tested the dice loss as the core loss and reported the results in Table 3. The best-performing version for dice loss $L_{G-NZ}$ for SM-D is consistently better than baseline and ARF-Net and has comparable results with ASP. This is a significant improvement as dice loss, to the best of our knowledge, has not been used alone for the mass segmentation due to training instability, yet using SM-loss greatly outperforms the hybrid setting.

