# OpenReview forum: "Spatial Matching Loss Function for Mass Segmentation on Whole Mammography Images"
_ICLR.cc/2024/Conference — ICLR 2024 Conference Withdrawn Submission_

### Official Review · Reviewer_EMHy · 2023-11-01

**Soundness:** 1 poor
**Presentation:** 1 poor
**Contribution:** 1 poor
**Rating:** 3
**Confidence:** 5

**Summary:**

The authors present a novel framework for calculating loss in the context of mass segmentation for mammography images. This approach incorporates higher-level spatial information by matching the spatial characteristics of predicted and ground truth masks during loss computation.

**Strengths:**

(1) The thoughtful design of the loss function is crucial for improving the performance of medical image segmentation models.

**Weaknesses:**

(1) The Spatial Matching (SM) loss is not sufficiently elucidated. In addition to providing algorithmic details and figures illustrating the method, a comprehensive description of the calculation process within the main text is warranted.
(2) Further experimentation with larger datasets is imperative to validate the method's robustness.
(3) It is essential to include distance metrics, such as the Hausdorff Distance (HD), in the evaluation.

**Questions:**

(1) While SM loss appears applicable to various medical image segmentation tasks, could you elaborate on its specific relevance and advantages in the context of mass segmentation?
(2) Could you provide a clear explanation for the significance of different colored cubes/circles within the proposed method?
(3) In the related work section, numerous segmentation losses are discussed. It would be beneficial to include comparative experiments to highlight the strengths and weaknesses of SM loss in relation to these existing methods.

---

> ### Author Response · Authors · 2023-11-23
>
> 1-We appreciate your feedback. Following the comment, we added more references to the Figures in the related parts. Included the meaning of the colors for Figure 1,2,4. Improved the captions. Inserted more explanation about the proposed method (pages 2, 4, 5, 6), notably in the following parts.
>
> 2- In general, for mass segmentation, INbreast and CBIS-DDSM are two publicly available benchmark datasets. INbreast has the precise segmentation masks that CBIS-DDSM lacks. In addition, INbreast is used in the literature with a common practice of the five-fold validation. We did not include the CBIS-DDSM for two reasons: first, the segmentation masks in all the images were not accurate. Secondly, to the best of our understanding, papers that utilized it for segmentation \cite{rajalakshmi2021deeply,xu2022arf,sun2020aunet} used different subsets, and it was not clear how the subsets were selected from the test and training for segmentation; therefore, a direct and fair comparison was not possible. We also tried DDSM-BCPR and MIAS. The segmentation masks for the DDSM-BCRP are like an approximation in the majority of the cases. The same reason holds for MIAS, in which the center of the mass with a radius is provided for a subset of samples. We will look for other datasets from this / other domains that are publicly available with accurate segmentation masks. However, after receiving the reviews, we conducted tests using Dice loss as the core loss and included the results in the appendix, which is consistently outperforming baseline, which is significant as dice loss, to the best of our knowledge, has not been used alone for the mass segmentation due to training instability, yet using SM-loss it outperforms the hybrid setting greatly.
>
>  3- We appreciate the comment. As with all the models we saved, we tested the models in the test set for each fold (the test set was recorded) and included the HAU in the experimental results. Added to table 1  and 2.
>
> 4- We added the following explanation about the relevance of SM-loss to mass segmentation to the paper.
>  In this study, we propose a spatial matching loss framework that incorporates
> higher-level information such as distance, size, and quantity in the calculation of a given core loss,
> aiming to alleviate the limitations of the previously proposed loss functions. This is specifically
> important for mass segmentation as spatial matching compares the masks locally and could alleviate
> the pixel class imbalance problem, which is a major problem in mass segmentation. In addition,
> due to the variety of sizes and appearance of the masses, specifically for samples with higher ACR
> density, distinguishing the mass from other regions of the image is more challenging, and the total
> mismatch between the location, quantity, and size of predicted and true masses is likely even in recent well-developed methods. Therefore, the proposed loss is beneficial for handling these challenges,
> which are common in mass segmentation in mammography images.
>
> 5- We apologize for omitting the explanation colored for the circles. The corresponding explanation was added to the revised paper in the caption of Figure 2 and in the text. The green oval shape is the mass, and the orange circle presents the predicted mass. We used colored cubes to explain which cubes were used in the scale-based and distance-based losses and referred to the colored cubes in the paper in the related subsections.
> The green (containing only true mass and not predicted mass) and yellow (containing both true and predicted mass) cells will be counted in the numerator of the scale-based version. Yellow and orange ( containing only predicted mass, not true masses) will be counted in the denominator for the scale-base loss Figure 2 (b-e). The distance will be calculated for the blue cells containing only predicted mass, not true mass, in the distance-based version.
>
> 6-In the related work section, numerous segmentation losses are discussed. It would be beneficial to include comparative experiments to highlight the strengths and weaknesses of SM loss in relation to these existing methods.
>
> 7-Considering this comment after receiving the reviews, we conducted experiments on Dice loss. In the revised version, we added the results of experiments for Dice loss, alongside BCE and Tversky, that were already presented in the paper; we tested the SM-loss on three different and most common losses for mass segmentation. In summary, Dice loss also performed considerably better using the SM framework. It would be noted that, to the best of our knowledge, Dice was not previously used alone as the loss for mass segmentation due to training instability; however, using the SM framework, dice loss outperformed the results of the common practice of using it in the hybrid setting with BCE loss.

---

### Official Review · Reviewer_SUy7 · 2023-11-01

**Soundness:** 3 good
**Presentation:** 2 fair
**Contribution:** 1 poor
**Rating:** 3
**Confidence:** 4

**Summary:**

This paper proposed a novel loss function, Spatial Matching (SM), for image segmentation applied to the problem of segmenting masses in mammography. The SM loss is based on evaluating a loss function over a grid within the mammogram image and then aggregating the local losses along with the overall image loss. An ablation study is performed to assess hyper-parameter choices and a comparison is made to existing state of the art methods on the INbreast dataset.

**Strengths:**

The paper identifies some of the limitations with conventional losses used such as binary cross entropy. Evaluation approach is appropriate for the problem at hand.

**Weaknesses:**

The paper is outlining a novel loss for segmentation motivated around the problem of mass segmentation in mammography. It could be argued that this is an artificial problem, in that the typical purpose of mass detection systems is to assist with the recall decision for screening mammograms either as an automated second reader or prompting system. Segmenting the masses itself is of less importance in clinical settings. No real attempt is made to outline why mass segmentation is of use.

As this paper focusses on the introduction of a novel segmentation loss, it would be helpful to see it evaluated on a range of datasets in other imaging domains to see if it has more broad utility.

Images are resized to 256x256 meaning that the algorithm processes very low resolution images. Does this have any effect detection/segmentation of small lesions?

Object segmentation approaches (i.e. Mask RCNN) are not considered in the background literature or evaluation.

Minor comments:

Figure 1 is hard to understand and doesn’t really do much to explain the limitations of BCE.

Figure 2 is also challenging to understand.

Figure 3 caption labels appear to be the wrong way round.

Figure 4 caption doesn’t which is ground truth and which is predicted.

No discussion of object detection approaches are taken (i.e. RCNN type).

Suggestions:

Add the colour codes to the figure captions so they can be understood independently.

**Questions:**

Please clarify why object segmentation type approaches were not considered. (i.e. Mask-RCNN)

---

> ### Author Response · Authors · 2023-11-23
>
> 1- The importance of mass segmentation is in the information that it provides, such as the exact location and the border of the mass, which are vital to detecting the abnormality type and assessing the performance of the method, specifically in samples with high ACR density that the detection of the mass is challenging. We added this to the introduction with references. It can be used for precise counter annotation generation as a mass segmentation approach was used to CBIS-DDSM \cite{lee2017curated} for precise segmentation mask generation for masses.
>
>
> 2-  While mass detection approaches Liu et al. (2021b;a) provide the bounding boxes of the
> mass, mass segmentation provides boundary information in addition to location and size, which is
> important for determining the abnormality type Guliato et al. (2007); Sun et al. (2020).
>
>
> 3- We appreciate the useful suggestion, which will be useful for our future work. For this paper, we focused on mass segmentation as it has all the challenges the SM-loss could help to alleviate these problems; however, this point is included in the paper in the introduction, and we will investigate it in the future.  However, after receiving the reviews, we conducted tests using Dice loss as the core loss and included the results in the appendix, which is consistently outperforming baseline, which is significant as dice loss, to the best of our knowledge, has not been used alone for the mass segmentation due to training instability, yet using SM-loss it outperforms the hybrid setting greatly.
>
>
> 4- We decided to use resize to (256 * 256) as it is a very common practice in the literature for mass segmentation in mammography images; most of the references in the paper for mass segmentation used the same size. The baseline method \cite{sun2020aunet} tested on larger the size of the images (512 * 512)  and reported a negative impact on the performance. Based on our observation, the majority of very small masses will still be segmented depending on different factors such as subtlety and ACR density category; it might become more challenging.
>
> 5- For the literature review for the mass segmentation part, we started with deep learning-based approaches for end-to-end pixel-to-pixel segmentation, starting with FCN and Unet (which is a common practice in most of the references in the paper) and continued to the ones for medical imaging and mass segmentation focusing on the variation of the U-Net as the baseline was in this category a and then moves to review losses. Mask-RNN is most commonly used to compare the mass detection approaches, and the ones that used it for segmentation on INbreast dataset \cite{min2020fully,soltani2021breast} either have heavy preprocessing or different experimental settings, so we were not able to include them in the evaluation and focused on pixel-level approaches that was directly related to the SM-loss.
>
> 6- We modified the caption of Figure 1 by adding more explanation. The limitations of BCE for mass segmentation are explained in detail on page 2 of the introduction.
>
> 7-Thank you for the comment. We added more references to Figure 2 in the body of the paper for each part. The caption is also improved.
>
> 8- The caption of Figure 3 was edited to match the order in the Figure. The explanation in the text already matched, so we did not change that part.
>
> 9- Thank you for the suggestion to improve the clarity of the paper. For Figure 4, we added the meaning of the colors as suggested. Green is the true mass, and blue presents the predicted mass.
>
>
> 10 - The main focus was on the approaches for pixel-to-pixel segmentation for SM-loss and AUnet; therefore, we focused on closely related approaches. In the future, we will try to include them in the comparison by experiments with the same setting as the main approach.
>
> 11-  Thank you for the suggestion to improve the clarity of the paper. We added the meaning of the colors in Figures 1,2,4 as suggested.
>
>
> 12- For the literature review, our main goal was to provide the reader with a background for pixel-level segmentation and the evolution around the Unet-type method and conclude with ones specializing in mass segmentation in mammograms. Therefore, within the limit of the available space for the paper, for the mass segmentation part, we started with UNet and FCN, which were among the very first pixel-level segmentation methods; we moved to more recent works focusing on the approaches for segmentation for medical images and finally mass segmentation for mammography images. Given the limited pages, we were not able to include another section for the current paper, but we will try to include it in future.

---

### Official Review · Reviewer_HK6n · 2023-11-06

**Soundness:** 3 good
**Presentation:** 3 good
**Contribution:** 3 good
**Rating:** 5
**Confidence:** 4

**Summary:**

The paper proposes a new loss function called spatial matching (SM) loss for mass segmentation on mammography images. The paper claims that spatial matching loss incorporates higher-level spatial information, such as distances, sizes, and quantities of the masses. The authors conduct mass segmentation experiments using the INbreast dataset, with AU-Net as the baseline, binary cross entropy loss, and Tversky loss as the core loss.

**Strengths:**

1. This paper goes beyond pixel-level segmentation losses and proposes a new way to incorporate high-level information. It creates a grid on the prediction and ground truth mask and aggregates the cell losses to reflect the mass distance, quantity and size information, which are important for mass segmentation in mammography images.
2. The authors give a good review of the recent mass segmentation methods for mammography images and loss functions for medical image segmentation and conduct experiments based on the recent mass segmentation network.

**Weaknesses:**

1. The paper’s main algorithm lacks explanation. In Section 2.3.2, the scale-based loss is to rescale the L_G with the scale ratio. However, how the definition of scale ratio can help overcome the size differences and the differences in the quantities of the masses is not explained at all. Thus, the paper’s argument to incorporate the size information is not supported.

2. The paper's contribution appears to be insufficient. The two methods proposed for aggregating cell losses seem straightforward. Moreover, the computation of the scale ratio in Section 2.3.2, which is based on the ratio of the number of cells, seems to achieve the same objective as simply computing the ratio of the number of pixels.

3. The paper contains some writing mistakes. Notably, the methodology section is incorrectly placed under "Related Work." Additionally, Eq. 5 and Eq. 8 are flawed, with Eq. 5 missing the indicator function in the numerator and Eq. 8 lacking summation in both the numerator and denominator. Furthermore, Fig. 1(b)'s caption mentions size and location differences, but the figure only depicts location differences. Lastly, Fig. 3(a) and (b) should be swapped.

4. The experiments conducted in the paper solely focus on one dataset, thereby inadequately supporting the claims made. Table 1 indicates that combining the scale loss and distance loss could lead to worse results compared to using either of them individually. In fact, with Tversky loss, the optimal outcome is achieved when the authors utilize the basic loss, which does not incorporate scale or distance.

**Questions:**

In Section 2.3.2, the scale ratio is defined as the number of cells containing positive pixels in the prediction over the number of cells containing the positive pixels in the ground truth. This would lead to high penalty when the prediction has much more positive regions than the ground truth, but lead to low penalty when the prediction has much less positive region than the ground truth. Does this make sense?

---

> ### Author Response · Authors · 2023-11-23
>
> 1-  We included more explanation in the introduction on pages 1 and 2. Also, additional explanations were added in the proposed method section on page 4.  The captions of Figures 1, 2, and 4 are clarified and referred to in related sections in the proposed method.
>
> 2- The numerator measures how large the predicted mass is relative to the true mass as the cell size is defined according to the true mass size, penalizing the large differences by increasing the loss. The cell size is a function of true mask size, so the denominator will be four at maximum for samples with one mass, which is the majority of samples. The values of more than
> one occur if the mass is divided between the cells. The sigmoid function helps to handle the small changes that can present small noise, like division up to four in the masses. One situation in which the denominator can have large values is when there are several masses in one sample. For these cases, as the cell size is selected based on the size of the smaller mass, the scale-based loss would not accurately present the initial idea of the scale, depending on the sizes of the true and predicted mass. The denominator might be considerably bigger than the numerator, which will decrease the impact of the grid-based loss depending on the magnitude of these samples, which is appropriate as the cell loss does not accurately present the true mass size. The scale implicitly
> penalizes the larger qualities of the predicted mass compared to the real mass, considering that if
> a certain number of predictions for the positive class are presented in one cell, they all will be
> counted as one cell. However, if the same number of positively predicted pixels are presented in
> five different parts of the image (more quantities), the number of cells for the same number of pixels
> will be larger; therefore, SM will penalize the difference in the quantity to some extent compared to
> pixel-level losses. It should noted the ratio of the number of pixels instead of the cells in the scale
> would not incorporate the quantities into the calculation.
>
>  3- We proposed SM loss, which is a very effective framework while being straightforward. In addition, the aggregation is one part of the proposed loss and not the total contribution of the paper,  even only that part provided a considerable boost to the performance of the baseline method, as shown in the result section. For instance, only the aggregation part of the paper surpasses AFR-Net, which is a more complicated design, highlighting the effectiveness of the SM loss while being simple in that it is able to overcome the challenges to a great degree.
>
> 4- This comment was answered in response number 2.
>
> 5-  The proposed method is modified and separated into section 3.
>
> 6-  Equation 8 is corrected. In Equation 5, the numerator is a summation of the cell losses, and the denominator counts the cells that contain at least one of the TP, FP, or FN.
>
> 7- By size difference, we were referring to the size difference between the foreground (mass) and the background. We clarified the meaning of "size difference" in the caption.
>
>  8- Caption of Fig 3 was corrected.
>
> 9-  For mass segmentation, INbreast and CBIS-DDSM are two public benchmark datasets. INbreast has the precise segmentation masks that CBIS-DDSM lacks. In addition, INbreast is used  commonly with five-fold validation. We did not include the CBIS-DDSM, as the segmentation masks in all the images were not accurate. Also to the best of our understanding, papers that utilized it for segmentation  (AU-Net, ARF-Net DSU-Net) used different subsets, and it was not clear how the subsets were selected ; hence, a fair comparison was not possible. We also tried DDSM-BCPR and MIAS. The segmentation masks for the DDSM-BCRP are largely inaccurate in the majority of the cases. The same reason holds for MIAS, in which the center of the mass with a radius is provided for a subset of samples. However, after receiving the reviews, we tested SM using Dice and included the results in the appendix, which is robustly outperforming baseline which is significant as dice, to the best of our knowledge, has not been used alone for the mass segmentation due to training instability, yet using SM-loss it outperforms the hybrid setting greatly.
>
> 10-  The effect of the scale and distance on a combination will vary in different settings, as some of them might have an overlapping effect; therefore, it is challenging to separate the contributions in the results. However, all the variations have a robust improvement on the baseline loss.
>
> 11-  This observation would have been problematic if the grid size was fixed for all samples, but the grill size was selected based on the size of the mass in each sample in this paper. Generally, the number of cells for the true mask would not exceed four as a result of the mismatch between the locations as cells, which we used the sigmoid to limit the effect of the small difference that might be this noise.

---

### Author Response · Authors · 2023-11-23
**General response**

Dear reviewers, we would like to thank you for reviewing our paper  "Spatial Matching Loss Function for Mass Segmentation on Whole Mammography Images" thoroughly and providing us with valuable comments to improve the presentations and the experimental section to highlight the strengths and weaknesses of the proposed method. In the following, all the concerns and questions have been addressed and clarified one by one. For the additional experiments, we conducted the experiment as much as the time allowed since receiving the reviews. In general, for comments that are suitable to include in the main body of the paper within the page limit, we included them in the paper. Those that were not possible to include in the main body are added to the appendix and referred to in the paper.
 The answers in individual responses are in the order of the comments in the reviews. For comments with several parts answers have been divided.

Due to the limitation in the responses we were not able to reference all the modification in the revised version, but all the explanations in the responses have been reflected in the revised version.

---

### Meta-Review · Area_Chair_CgcG · 2023-12-06

**Metareview:**

This paper presents a new loss for mass segmentation in nomography, which aims to address the limitation of BCE loss without considering the distance between predicted and true masses. The strength of the work is that the idea seems to be well motivated. However, the issue is a generic issue and the experimental validation is done in a specific medical image dataset. It would be greatly improved if the authors conducted some experiments in various applications and justified the generality of the method.

The presentation of the paper could also be improved to make the paper easy to understand. For example, the authors may use real-world examples instead of toy samples in Figure 1. The experimental results shall highlight how the proposed loss compared with previous commonly used losses, and in various different applications.

**Justification For Why Not Higher Score:**

The paper faces several major weakness and shall be improved.

**Justification For Why Not Lower Score:**

NA

---

### Decision · Program_Chairs · 2024-01-16

Reject